# Plant Cyanogenic-Derived Metabolites and Herbivore Counter-Defences

**DOI:** 10.3390/plants13091239

**Published:** 2024-04-29

**Authors:** Manuel Martinez, Isabel Diaz

**Affiliations:** 1Centro de Biotecnologia y Genomica de Plantas, Universidad Politécnica de Madrid (UPM)—Instituto Nacional de Investigación y Tecnología Agraria y Alimentaria (INIA/CSIC), Campus de Montegancedo, Pozuelo de Alarcón, 28223 Madrid, Spain; m.martinez@upm.es; 2Departamento de Biotecnologia-Biologia Vegetal, Escuela Técnica Superior de Ingeniería Agronómica, Alimentaria y de Biosistemas, Universidad Politecnica de Madrid, 28040 Madrid, Spain

**Keywords:** cyanide, cyanogenesis, detoxifying mechanisms, herbivore adaptation

## Abstract

The release of cyanide from cyanogenic precursors is the central core of the plant defences based on the cyanogenesis process. Although cyanide is formed as a coproduct of some metabolic routes, its production is mostly due to the degradation of cyanohydrins originating from cyanogenic glycosides in cyanogenic plants and the 4-OH-ICN route in Brassicaceae. Cyanohydrins are then hydrolysed in a reversible reaction generating cyanide, being both, cyanohydrins and cyanide, toxic compounds with potential defensive properties against pests and pathogens. Based on the production of cyanogenic-derived molecules in response to the damage caused by herbivore infestation, in this review, we compile the actual knowledge of plant cyanogenic events in the plant–pest context. Besides the defensive potential, the mode of action, and the targets of the cyanogenic compounds to combat phytophagous insects and acari, special attention has been paid to arthropod responses and the strategies to overcome the impact of cyanogenesis. Physiological and behavioural adaptations, as well as cyanide detoxification by β-cyanoalanine synthases, rhodaneses, and cyanases are common ways of phytophagous arthropods defences against the cyanide produced by plants. Much experimental work is needed to further understand the complexities and specificities of the defence–counter-defence system to be applied in breeding programs.

## 1. Introduction

In the more than 400 million years of evolution shared by plants and phytophagous arthropods, both adversaries have developed physical and chemical defence strategies and have adapted their physiology and behaviour habits to protect against each other [1]. Derived from this long coexistence, plants have settled a precise perception and recognition of the feeder. Recognition is translated into gene reprogramming and the synthesis of a plethora of selected direct and indirect chemical and physical defences, with a high degree of specificity [2,3,4,5]. Chemical defences include molecules with repellent, anti-nutritional, deterrent, or toxic properties to interfere with the aggressor metabolism and its growth, along with the release of volatiles to attract natural enemies as collaborators in the battle [6,7]. Therefore, plants have evolved and gained the ability to detect herbivore attacks through specific plant receptors (PRRs; pattern recognition receptors) and trigger defence strategies that are dependent on their physiology and are specifically produced to combat each herbivore species with a precise feeding mode. Thus, plants efficiently induce physical and chemical barriers against herbivores, which, in turn, counteract to restrict these defences and to overcome, diminish, or adapt to detrimental effects. In consequence, plants react and, as a second defensive mechanism, counter-attack and implement emergency responses [3,8].

Cyanogenic plants are characterised by their ability to release hydrogen cyanide (HCN). These particular plants produce cyanogenic compounds, mainly cyanogenic glycosides (CNglcs), with potential defensive properties against herbivores, pathogens, and even mammals and birds [9,10]. CNglc is α-hydroxynotrile stabilised with sugar [11]. To be effective, CNglcs require two enzymatic steps. First, they are de-glycosylated by β-glucosidases to separate the sugar moiety and liberate cyanohydrins (α-hydroxynitriles). Cyanohydrins are then hydrolysed in a reversible reaction catalysed by an α-hydroxynitrile lyase (HNL), generating carbonyl compounds (aldehyde or ketone) and HCN (Figure 1; [11,12]). This two-component system to transform protoxin into a toxic molecule is not exclusive to cyanogenic plants; it is also used for the production of other plant defensive metabolites such as glucosinolates and benzoxazinoid and iridoid glycosides [13,14,15]. The two-step adaptive mechanism has probably evolved to reduce autotoxicity, since substrates and enzymes are compartmentally separated, and only bring them together when the plant cell/tissue disruption is either mediated by herbivore feeding or by physical damage.

Previous studies have revealed that CNglcs are non-toxic forms stored in vacuoles, while β-glucosidases accumulate in the apoplast bound to the cell wall of dicot plants, and in the chloroplast or the cytoplasm in monocot species [16]. HNLs have been localised in the cytoplasm and the plasma membrane [12]. In other plants such as sorghum, this separation occurs at the tissue level, being the CNglc dhurrin accumulated in the leaf epidermal layer and β-glucosidases and HNL in leaf mesophyll cells [17]. Consequently, the catalytic reaction and the release of harmful secondary metabolites are only produced in response to the injury, being these molecules directly targeted to the feeder physiology. 

In brief, plant cyanogenesis is defined as the release of HCN, and the central core of this process is the reversible transformation of cyanohydrins into cyanide, being both toxic defensive molecules [11]. Based on this concept and having a scenario where cyanogenic-derived molecules are produced in response to the damage caused by herbivore infestation, in this review, we have tried to compile the actual knowledge of plant cyanogenic events in the plant–pest context. Besides the defensive potential, the mode of action, and the targets of the cyanogenic compounds to combat phytophagous insects and acari, special attention has been paid to arthropod responses and the strategies to overcome the impact of cyanogenesis.

## 2. Cyanogenesis in Plants

### 2.1. Cyanogenic Plants

Over 3000 species are considered cyanogenic plants, including angiosperms, gymnosperms, and ferns, since they synthesise cyanogenic compounds. As these metabolites have not been detected in algae and bryophytes, cyanogenesis was initially catalogued as a common chemical defence trait that appeared in vascular plants more than 300 million years ago [18]. Now, it is considered a feature that has evolved independently in several plant lineages through the recruitment of members from similar gene families [19]. An example is found in the genus *Eucalyptus*, where different species have recruited UDP-Glycosyl-transferases (UGTs) from different families for prunasin biosynthesis [20]. Selection pressures for the gain/loss of cyanogenic capabilities are far to be known. The amount of nitrogen in soils has been associated with the cost of forming these nitrogen-containing specialised metabolites, and thus, it has been considered an important factor driving selection for or against the presence of cyanogenic glucosides [21]. 

A recent overview by Yulvianti and Zidorn [22] has discussed the chemical diversity of 112 CNglcs already found in plants with structures that vary between plant species and depend on the amino acid from which they derive. Among CNglcs resultant from aliphatic amino acids, the most widely studied are linamarin and lotaustralin of which their precursors are Val and Ile, respectively. They have been mainly described in lima beans (*Phaseolus lunatus*), cassavas (*Manihot esculenta*), and some forage plants [11]. Additionally, dhurrin derived from Tyr, and prunasin and amygdalin derived from Phe, are also well known and have been mostly linked to *Sorghum* and *Prunus* genera, respectively [23,24].

Apart from their defensive action, CNglcs also serve as a nitrogen source for amino acids and other N-containing molecules and participate in several physiological processes since their mobilisation does not always imply the release of HCN. CNglcs have been involved in plant growth and development and act as osmoprotectants and scavengers of reactive oxygen species associated with plant responses to abiotic stresses [25,26]. A recycling turnover pathway for CNglcs has also been defined by which glycosides of amides, carboxylic acids, and “anitriles” appear as common intermediates with biological significance for the plant [27]. However, the main function of CNglcs in plants is probably due to the production of cyanide and its action as a modulator and signalling molecule, besides its toxic effect [28].

CNglcs derive from aromatic, aliphatic, and, in some cases, non-proteogenic amino acids and require three enzymatic reactions to be consecutively converted into oximes and α-hydroxynitriles and finally glycosylated by UDP-Glycosyl-transferases (UGT85B/K) to form CNglcs (Figure 2) and some related non-cyanogenic glycosides termed rhodiocyanosides [10,23]. The enzymes involved in the two first catalytic reactions belong to the cytochromes P450. In particular, the CYP79 family (CYP79D1/D2/A1) catalyses the formation of oximes, and the CYP71, CYP736, CYP706 and CYP38 families participate in their subsequent conversion into α-hydroxynitriles [10]. The apparent simplicity of this synthetic pathway and the functional efficiency of the enzymes involved were validated by the conversion of the non-cyanogenic tobacco and *Arabidopsis* plants into cyanogenic species. Both plants were able to accumulate dhurrin after being stably transformed with *CYP79A1* and *CYP71E1* genes from sorghum [29]. Moreover, the use of radiolabelled Tyr as the initial precursor of the reactions demonstrated that the synthesis of dhurrin was dependent on the expression of both sorghum P450s. Likewise, the simultaneous expression of *CYP79D71* gene from *P*. *lunatus* and *CYP736A2* and *UGT85K3* genes from *Lotus japonicus* resulted in the production of linamarin and lotaustralin in infiltrated *Nicotiana benthamiana* leaves [30]. Alternatively, the generation of loss-of-function cassava plants either via the *CYP79D1/D2* anti-sense approach and, more recently, through the targeted mutagenesis of the *CYP79D1* gene mediated by CRISPR/cas9 genome editing, highly reduced linamarin levels in tobacco leaves [31,32]. These data demonstrate the potential of these genetic approaches to control the amount of certain specific metabolites linked to cyanogenic plants and the feasibility of expressing cyanogenic compounds in non-cyanogenic plants to improve defences against pests.

Besides HCN’s role as a poison, two very recent reviews have highlighted its function as a gasotransmitter-signalling molecule independent of its toxicity, essential in several physiological processes [28,33]. Its mode of action includes the post-translational modifications of proteins at the cysteine -SH groups which have effects on the protein folding, stability, subcellular location, and consequently, their activity. As Diaz-Rueda et al. indicate [28], cyanide signalling is of special relevance since it triggers fast and efficient responses to restore homeostasis and/or adapting plant physiology to changing environments and to the emergence of new pests [28].

As mentioned above, to be effective and generate cyanohydrins and/or cyanides as defences against biotic stresses, CNglcs require specific β-glucosidases to separate sugar moiety and liberate cyanohydrins. Thus, the β-glucosidase activity determines the kinetic of cyanide release and, consequently, the plant toxicity [34]. The relevance of this catalytic process was shown by the search of mutants in the catabolism of CNglcs in the legume *L*. *japonicus* performed by Takos et al. [35]. Among a set of mutants, they identified the *cyd2* mutant deficient in the β-glucosidase BGD2 and demonstrated that its absence eliminated cyanogenesis in *Lotus* leaves. A very closely related gene in white clover (*Trifolium repens*) is the *Li* gene, also encoding a β-glucosidase involved in CNglc breakdown. This gene was present in cyanogenic clover accessions but not in the non-cyanogenic ones [36]. The role of the β-glucosidases in cyanogenesis was corroborated by incubating leaf extracts of *N*. *benthamiana* transiently transformed with the *L*. *japonicus BGD3* glucosidase gene with exogenous prunasin, lotaustralin, and linamarin as substrates [37]. These authors detected a significant reduction in the starting level of substrates after 30 min of incubation. Additionally, transferring the complete pathway of *Sorghum bicolor* for the synthesis of dhurrin together with the corresponding specific β-glucosidase into *A*. *thaliana*, converted this species into cyanogenic and provided it resistance against the coleopteran *Phytollotreta memorum* [38]. These experiments with transformed plants validated the significance of the glucosidases in the cyanogenic events and underlined the effectiveness of cyanogenesis as an anti-herbivore defence system.

The costs and benefits of HCN and cyanohydrins against herbivores vary with the availability of resources, herbivore pressure, and plant functional traits [39]. These secondary metabolites with defensive properties seem to be part of a cost-effective strategy since they can be temporarily stored and, upon demand, remobilised, reallocated, or transformed in metabolites with a plant growth or defence function. For example, the CNglc linamarin accumulated in *Hevea brassilensis* seeds is glycosylated following germination and transported to cotyledons where they can be stored as potential defence products to be catabolised, producing cyanohydrins and HCN. Alternatively, it can be remobilised from cotyledons into leaves and roots and, after some metabolic reactions, participate in latex generation and rubber production [21]. So, the rubber tree linamarin, as well as other CNglcs from other plant species, can be used as a supplementary source of nutrients if they are required for plant growth or be sequestered, maintaining a level of chemical defence to be used as substrates for the synthesis and release of defence compounds. Thus, the benefits of these anti-herbivore defences depend on the degree to which a defence favours plant fitness in the presence of a feeder.

### 2.2. Cyanohydrins: Convergent Molecules of Cyanogenic and Non-Cyanogenic Plants

The production of cyanohydrins is not exclusive to cyanogenic plants, the Brassicaceae family, of which its members are considered non-cyanogenic species, possess an alternative Trp-derived pathway to synthesise cyanohydrins. Brasicales also have HNL enzymes to hydrolyse cyanohydrins releasing HCN, as cyanogenic plants do. This metabolic route described in *A*. *thaliana*, known as the 4-OH-ICN pathway, redirects the oximes derived from Trp to produce an alternative cyanogenic indole carbonyl nitrile (ICN) [10,40]. 

The 4-OH-ICN pathway shares with the CNglc biosynthetic route the initial step, which transforms Trp into indole-3-acetaldoxime (IAOx) catalysed by the redundant CYP79B2 and CYP79B3 monooxygenases. This reaction is also a common starting point for the synthesis of camalexin and indole glucosinolates (Figure 2), all important secondary metabolites associated with the defence of Brassicaceae plants [41,42]. Regarding the particular 4OH-ICN pathway in *Arabidopsis*, IAOx is transformed in an indole cyanohydrin, then to ICN, and finally into cyanohydrins by the consecutive action of CYP71A12, the Flavin-dependent oxidoreductase FOX1 and the CYP82C2 enzymes [40]. The four abovementioned genes (*CYP79B2*, *CYP71A12*, *FOX1*, and *CYP82C2*) are necessary for the complete reconstitution of the 4-OH-ICN pathway in agroinfiltrated *N*. *benthamiana* plants as demonstrated by the accumulation of 4-OH-ICN derivatives, in particular, the 4-OH-indole-3-carbonyl methyl ester, in their leaves [39]. This metabolic pathway exclusively found in Brassicaceae is considered a metabolic reinvention to expand plant defences. Its effectiveness in providing plant resistance was first proven against bacterial and fungal pathogens and more recently to herbivores [10,40,43]. In this context, it is important to highlight a recent work published by Arnaiz et al. [12] who reported the upregulation of a set of genes encoding the enzymes of the 4-OH-ICN pathway, such as CYP71A12, FOX, β-glucosidase, and HNL, in spider mite-infested *Arabidopsis* in comparison to non-infested plants, and detected an increase in HCN and cyanohydrin content to act as defensive compounds.

### 2.3. The Central Core of Cyanogenesis: The Reversible Transformation of Cyanohydrins into Cyanide

The release of cyanide from cyanogenic precursors that occurs upon tissue damage is the central core of the cyanogenic defences (Figure 1; [11]). This reaction, involved in the last step of cyanogenesis, may happen spontaneously or be catalysed by the HNL and is essential from the defensive point of view. Indeed, *Arabidopsis HNL*-overexpressing plants infested by spider mites presented less damage than mite-infested wild-type plants and much less than infested *nhl* mutant lines. The reduced damage was concomitant to an increase in cyanohydrin or HCN content and a restriction in mite fecundity after feeding on those plants. Thus, Arabidopsis *HNL*-overexpressing plants were more resistant to spider mite *Tetranychus urticae* attacks [12]. 

*HNL* genes have been identified in cyanogenic and non-cyanogenic plants and their reversible character has been proposed as a strategy to maintain non-toxic levels of HCN within the plant [12,44]. In addition, there is a group of HNLs described in some cyanogenic species that catalyse the dissociation of the cyanohydrin known as mandelonitrile to HCN and benzaldehyde. They have been mainly studied in *Prunus* genus, focusing on their association with seed physiology [45,46]. However, their potential defensive role is still unknown.

HCN does not only result from the two mentioned metabolic routes but is also naturally produced in plants as a coproduct of other metabolic routes (Figure 2). Ethylene synthesis is a branch from the adjacent methionine Yang cycle and forms cyanide by the oxidation of the 1-amonocyclopropane-1-carboxylic acid (ACC) during its conversion to ethylene [47]. Likewise, indole-3-acetonitriles (IANs), intermediates derived from Trp, may be converted into camalexin and also release HCN. In plant tissues, hydrogen cyanide is also formed from glyoxylate, a photorespiration product, and from hydroxylamine, an intermediate of nitrate assimilation [33,48]. Cyanide is additionally liberated from glucosinolate breakdown which also produces isothiocyanates, thiocyanates, epithionitriles, and nitriles, all these metabolites having toxic or deterrent properties against herbivores [48,49]. Cyanolipids, as cyanohydrin esters, are another source of HCN [50]. Thus, cyanide is a ubiquitous plant-derived compound that originates from many metabolic pathways (Figure 2). 

## 3. Toxicity of Cyanogenic-Derived Compounds

Cyanide and cyanohydrins are toxic molecules, particularly the HCN with a mechanism of action and targets that are well known. The HCN, also termed prussic acid, is a potent poison that is highly reactive and able to inhibit mitochondrial oxygen respiration. It is not accumulated as a free form but is released from cyanogenic precursors upon tissue damage. Cyanide liberation takes place upon herbivore attack and conforms to the concept of ‘cyanide bomb’, which is widespread in the plant kingdom [51,52]. HCN toxicity is based on its binding affinity for the ferric-heme a_3_ form of cytochrome *c* oxidase, the final enzyme in the complex IV of the electron transport system. Cyanide acts as a non-competitive inhibitor of cytochrome c, and the cytochrome oxidase-CN complex blocks mitochondrial electron transfer, leading to the termination of the respiratory chain, cytotoxic hypoxia, and cell death [52]. HCN binds also to metal ions of some metallo-enzymes, and forms Schiff base intermediates provoking non-efficient metabolic reactions. Additionally, it can interfere with the activity of enzymes linked to the redox homeostasis via the S-cyanylation or oxidation of proteins and has the ability to modify cysteine-containing proteins [48]. 

The dual role of cyanide either a toxin or a regulatory molecule depends on Its concentration in plant tissues. At high toxic levels (over 100 µM to millimolar), it may be used in defence against herbivores [48]. This role is well documented and supported by arthropod feeding bioassays with transgenic plants [10,23,39]. In contrast, low HCN concentrations (1–100 µM) may have a regulatory function, mainly signalling, participating in many different physiological events [28,48]. So, the content of cyanide in plants requires a fine-tuned balance for the development of its defensive and regulatory roles as well as for keeping food safety which is tightly regulated by enzymatic pathways and detoxifying enzymes to re-metabolise cyanide [53]. The HCN impact on arthropods depends on doses and exposition time and differs between species. Cyanide interferes with mitochondrial respiration and glycolysis and increases proteolytic pathways [52]. Together with its toxic features, HCN possesses a bitter taste, leading to the deterrence of feeding in some insects [11]. Thus, its defensive role may be directed to a range of targets associated with the physiology and behaviour of arthropod feeders. Similarly, cyanohydrins and derivatives are also harmful molecules that easily penetrate lipoid insect tissues where either by themselves or by their potential to produce HCN have defence properties against pests. Thus, some cyanohydrins and cyanohydrin esters, either derived from plants or with a synthetic origin have been proven to have a valuable action as fumigants to pest control [54,55,56]. 

## 4. Herbivore Counter-Defences to Overcome Plant Cyanogenic Strategies

### 4.1. Herbivore Adaptation to Cyanogenic Defences

HCN is always present in plants although some cyanogenic species, and Brassicales have higher potential to produce greater concentrations of this metabolite. In general, cyanogenesis is considered an effective defence strategy against generalist herbivores, and it seems to have a minor impact on specialists. Herbivores with a large range of hosts may select or combine different food resources and feed on cyanide-rich or cyanide-free diets. In contrast, plant feeders with a restricted host number, termed specialist or oligophagous, have less choice and, consequently, have evolved physiological and behavioural adaptations to tolerate or detoxify harmful compounds [52]. Evidence of herbivore counter-adaptations has been found in Pierids. The association of Pierid butterflies and plants is an illustrative example of the ‘coevolutionary arms race’ adaptive process. The ability of Pierid species to safely handle cyanide contributed to the primary host shift from Fabales to Brassicales that occurred about 75 million years ago and was followed by Pierid species diversification. The key evolutionary innovation to colonise glucosinolate-containing plants was identified as a gut nitrile-specifier protein that redirects glucosinolate hydrolysis to nitriles instead of the toxic isothiocyanates. These nitriles decompose spontaneously into an aldehyde and cyanide, turning the ‘mustard oil bomb’ into a ‘cyanide bomb’ inside the larvae [57].

Cyanogenesis implies that plant tissue disintegration is required for HCN release. In their evolution, phytophagous arthropods have evolved different adaptive strategies to evade the spatial mixture of substrates and enzymes and prevent cyanide production. A general adaptive mechanism within arthropods to avoid defensive metabolites is based on their feeding habits. Thus, spider mites and aphids possess specialised sucking stylets to access plant nutrients from mesophyll cells or phloem sap, respectively. Their stylets are inserted through apoplast into the sieve elements (aphids), between epidermal cells or via opening stomata (mites) to avoid disturbing cells/tissues, and the consequent rapid release of plant defensive molecules [58]. In contrast, chewing species destroy plant cells/tissues to obtain nutrients, and consequently, it becomes more difficult for them to overcome defences. Nevertheless, some herbivores, mainly specialists, have deployed countermeasures either to avoid or to use these compounds for their own benefit and defence [59]. A clear example of adaptive feeding behaviour is the case of some lepidopteran larvae with a leaf-snipping mode that minimises tissue damage by leaving big portions of intact tissues and increasing the feeding speed to shorter reaction times for HCN production [52,60]. Alternatively, to keep CNglcs and hydrolytic enzymes spatially separated, some coleopteran, hemipteran, and lepidopteran species selectively sequester CNglcs from host plants and store them in their bodies. The end is to keep CNglcs far from the plant β-glucosidases retained in the insect saliva and gut lumen [61]. This is the case of the lepidopteran *Zygaena filipendulae* and *Heliconius melpomene* that sequester CNglcs but also synthesise them de novo to further be used for HCN emission as a defence mechanism against their natural enemies [52,61]. Another plan of counter-defence developed by lepidopteran herbivores to escape the toxic effects of cyanogenic-derived metabolites is the control of β-glucosidase levels in their mid-guts to reduce CNglc breakdown [62]. Physiological conditions of the herbivore guts together with specific sets of digestive enzymes determine the destiny of cyanogenic ingested compounds. The highly alkaline pH in the mid-gut lumen of numerous larvae of lepidopteran species inhibits the β-glucosidase action and contributes to keeping CNglcs intact [60,61]. Thus, either the avoidance of contact between substrates and enzymes or the inhibition of enzyme activity during the plant–arthropod interaction provokes the presence of intact CNglcs in the faeces of many lepidopteran species, with being secretion an efficient widespread mode of circumventing toxicity [61].

In general, to be more efficient and overcome plant defences, many arthropods combine several of the mentioned adaptation approaches during plant feeding and digestion. For example, larvae from *Z*. *filipendulae* that feed preferentially on the cyanogenic *L*. *corniculatus*, sequester linamarin and lotaustralin from this legume when is used as a food source. At the same time, larvae with a leaf-snipping feeding mode cause minimal tissue damage, and the highly alkaline pH of their digestive tract inhibits plant β-glucosidase activity to avoid CNglc de-glycosylation [60].

Unexpectedly, some species of the Lepidoptera order, such as *Spodoptera eridania*, grow better when cyanide takes part of their diets [63]. This response depends on the HCN levels since, at low concentrations, cyanide stimulates larva feeding. Larvae of this lepidopteran presented high tolerance to cyanide due to a certain insensitivity of the larvae cytochrome c oxidase involved in the mitochondrial electron transport system. Heisler et al. [64] demonstrated that mitochondria from bodies of *S*. *eridania* larvae presented low sensitivity to inhibition by KCN. This target-site insensitivity in the armyworm most likely depends on direct enzymatic detoxification and, in part, contributes to a high degree of cyanide tolerance. Indeed, a wide variation in the sensitivity among insects at different metamorphosis stages has been documented [64]. However, much experimental work is needed to confirm this hypothesis. If this is so, increasing the tolerance of these oxidases could be an additional adaptation mechanism to overcome cyanogenic-derived toxic products.

### 4.2. Herbivore Detoxification of Cyanogenic-Derived Compounds

In addition to physiological and behavioural adaptations, a common way phytophagous arthropods defend against the cyanide produced by plants is the detoxification of this toxic compound. Three families of enzymes active in cyanide detoxification have been reported. These families comprise β-cyanoalanine synthases (CAS), rhodaneses, and cyanases (Figure 3). In addition, chemical modifications of plant CNglcs during the feeding of some arthropods have been described.

#### 4.2.1. β-Cyanoalanine Synthases (CAS)

CAS catalyse the substitution of the sulfhydryl group of cysteine by cyanide, producing β-cyanoalanine and sulphide (Figure 3). β-cyanoalanine is also a deterrent to herbivores but is converted into Asn, Asp, and NH_4_ by specific nitrilases [23]. CAS enzymes have been broadly described in bacteria and plants [65]. In metazoans, the presence of CAS enzymes is restricted to some taxonomic groups, mainly to arthropod species, and originated from horizontal transfer events from bacteria [66,67]. The CAS enzymes of many arthropods have a bifunctional role, consisting of cysteine synthase and β-cyanoalanine synthase activities [68]. The cysteine synthase catalyses the production of cysteine from O-acetylserine and sulphide, which neutralises the potential toxic activity of sulphide in the mitochondria [69].

CAS seems relevant to protecting arthropod herbivores feeding on cyanide-defended plants as CAS activity is broadly distributed in these organisms [68,70]. CAS activity was demonstrated as labelled CAS was detected after feeding or exposition of herbivores to isotopically labelled cyanide [57]. An enzyme from the two-spotted spider mite *T*. *urticae* was the first CAS cloned and characterised [65]. Interestingly, mites acquired this gene from bacterial symbionts by an ancient horizontal gene transfer event [66]. TuCAS is more efficient in the synthesis of β-cyanoalanine than in the biosynthesis of cysteine and can detoxify cyanide using an alternative substrate, O-Acetyl-L-serine [67]. Structural data and sequence comparisons suggest that enzyme–substrate preferences may be controlled by molecule regions located further from the active site [67]. The effect of cyanide detoxification by CAS on mite performance was demonstrated by Arnaiz et al. [12]. The delivery of dsRNA-TuCAS to mites caused a significant reduction in the fecundity of treated mites after feeding on *Arabidopsis* plants [12]. Furthermore, the ability of mites to adapt to counter cyanide toxicity was found to be dependent on CAS activity [71]. Lepidopteran genomes contain sequences that group with the CAS sequence from *T*. *urticae* in phylogenetic analyses, suggesting a similar origin [66,72]. In many lepidopteran insects, CAS genes have undergone duplication following horizontal transfer. CAS duplications have been detected in some lepidopteran species that feed on cyanogenic host plants such as *H*. *melpomene*, *Spodoptera litura*, and *Pieris rapae* [70,73,74]. Duplicated CAS genes show marked divergence in gene expression patterns and enzymatic properties [72], which could have facilitated the adaptation of lepidopteran insects to different diets.

#### 4.2.2. Rhodaneses

Rhodaneses (thiosulfate sulfurtransferases) catalyse the transfer of sulphur from thiosulfate to cyanide, leading to the formation of thiocyanate and sulphite (Figure 3). Rhodaneses are widely distributed in both prokaryotes and eukaryotes, where they play a relevant role in mitochondrial function [75]. The presence of isotopically labelled thiocyanate in *P*. *rapae* larvae treated with isotopically labelled HCN indicated that larvae also can detoxify cyanide by rhodaneses [57].

A minor role of insect rhodaneses in cyanide detoxification has been proposed, based on the broad distribution among insects and their similar activity levels among herbivores that frequently or rarely encounter high cyanide levels [52]. The basal rhodanese activity might be sufficient to capture dietary cyanide in herbivores regardless of the cyanide level in the diet. Alternatively, a general role of rhodaneses in the regulation of sulphur homeostasis has been proposed [76]. For some rhodanese-like proteins, a higher affinity for mercaptopyruvate than for thiosulfate has been demonstrated, identifying them as mercaptopyruvate sulphur transferases. These proteins share high sequence identities with rhodaneses and can also transfer sulphur to cyanide [77]. To date, only three rhodaneses from insects have been characterised. In *Musca domestica*, the rhodanese MdRDH1 confers oxidative stress tolerance and participates in immunity [78]. In *P*. *rapae*, two rhodaneses (PrTST1 and PrTST2) were identified [79]. Based on the different kinetic properties and predicted subcellular localisation of PrTST1 and PrTST2, different physiological functions were proposed. PrTST1 could have a role in cyanide detoxification as it is presumably located in the mitochondria and has a much higher affinity for cyanide than PrTST2. The very low affinity for cyanide of PrTST2, a predicted cytosolic enzyme, suggests alternative functions. Expansions in the rhodanese family have been associated with specific roles acquired by Pierid species in the course of host plant adaptation [79]. However, a positive effect of cyanide detoxification by rhodaneses on insect performance has not been demonstrated yet.

#### 4.2.3. Cyanases

Cyanate metabolism relies on the well-characterised enzyme cyanase, which catalyses the reaction of cyanate with bicarbonate to produce ammonium and carbon dioxide (Figure 3). Cyanases require the previous oxidation of cyanide into cyanate catalysed by cyanide monoxygenases. The cyanase gene has been identified in many species and can play a significant role in the assimilation of exogenous cyanate and the detoxification of internally generated cyanate [80]. Interestingly, horizontal gene transfer (HGT) of cyanase genes has been widely found and contributed to the evolution of eukaryotes, including metazoans. Examples of multiple HGT were found in nematodes, where cyanases were partitioned among bacterial and plant sources [81,82].

Among arthropods, a horizontally acquired cyanase enzyme has been characterised in *T*. *urticae* [83,84]. This cyanase gene is transcribed in all mite-feeding stages (larvae, nymphs, and female adults) and is functionally active. It metabolises cyanate with similar kinetics as what has been reported for the plant and fungal eukaryotic enzymes. The genomes of some non-phytophagous mites also possess cyanase genes, which raises the question of whether cyanide detoxification is the main function of *T*. *urticae* cyanase [85]. Cyanate is also formed by the dissociation of carbamoyl phosphate, a main substrate for arginine and pyrimidines biosynthesis. Therefore, cyanase might be involved in the regulation of arginine and pyrimidine biosynthesis by changing cyanate concentrations [86]. Whether cyanase exerts a positive effect on mite performance based on cyanide detoxification has not been demonstrated.

#### 4.2.4. Phosphorylation and Glucosylation of CNglcs

The detoxification of xenobiotic compounds may also be reached by chemical modifications. The phosphorylation of toxic compounds occurs widely in insects, which supports its potential for detoxification [87]. Likewise, transglucosidation has been proposed as a common detoxification mechanism in insects. In the whitefly *Bemisia tabaci*, the hydrolytic activation of glucosinolates during feeding is prevented by the addition of glucose moieties via a transglucosidation mechanism [88]. The detoxification of cyanogenic glucosides has also been described by conversion to non-activable derivatives [89]. During plant feeding, the hemipteran *B*. *tabaci* produces the cyanide detoxification product β-cyanoalanine. To avoid cyanide production, the plant CNglc linamarin can be glucosylated and/or phosphorylated during arthropod feeding. Chemical modifications of linamarin impede the activation of this CNglc by the plant enzyme linamarase, avoiding cyanogenesis.

## 5. Conclusions and Future Perspectives

Defence–counter-defence events are commonly observed in the interaction between plants and phytophagous arthropods. Cyanide-based affairs are excellent examples of plant and arthropod creativity. Plants expand their defensive arsenal, avoiding auto-toxicity. Cyanogenic plants store cyanide as non-toxic cyanogenic glycoside molecules. Tissue disruption caused by chewing or sucking herbivores brings together CNglcs and their degrading enzymes, resulting in cyanohydrin and cyanide production. Brassicaceae have developed a specific route for cyanohydrin–cyanide generation triggered by herbivory. In response, herbivores develop diverse ways to avoid cyanide toxicity. Some strategies seek to evade the spatial mixture of substrates and enzymes to prevent cyanide production. Mites and aphids avoid disturbing cells/tissues and the consequent rapid triggering of plant defence. Some insects selectively sequester CNglcs from host plants and store them in their bodies. The highly alkaline pH in the midgut lumen of lepidopteran species inhibits the β-glucosidases activity and contributes to keeping CNglcs intact. In addition, many phytophagous arthropods can detoxify cyanide using enzymes coming from horizontal gene transfer events from bacteria, such are β-cyanoalanine synthases and cyanases. Furthermore, enzymes potentially used to detoxify the cyanide produced in the herbivore, such as rhodaneses, may be also used to detoxify xenobiotic cyanide. All mentioned events associated with cyanide production in the plant–herbivore context are schematically summarised in Figure 4.

The detected complexity of the cyanide-based interactions between plants and herbivores makes it difficult to predict the effectiveness of the produced cyanide and cyanohydrins as herbivore deterrents. Besides the concentration of cyanide precursors in the plant and its capacity to release cyanide, the mechanisms developed by the herbivore to counteract cyanide toxicity determine the success of plant defence. Further advances in the knowledge of the defence–counter-defence mechanisms are necessary to develop new strategies based on the capacity of cyanide and cyanohydrins to improve the control of phytophagous pests.

In summary, cyanogenesis plays a central role in plant defense to biotic stresses. Additional information on how environmental conditions regulate this process will aid to a better understanding of the future evolution of cyanogenesis in the plant–herbivore context. Further knowledge of the biosynthesis, bioactivation, and remobilisation events during plant development linked to cyanogenic-derived products could improve the resilience, efficiency, and yield of cyanogenic plants.

## Figures and Tables

**Figure 1 plants-13-01239-f001:**
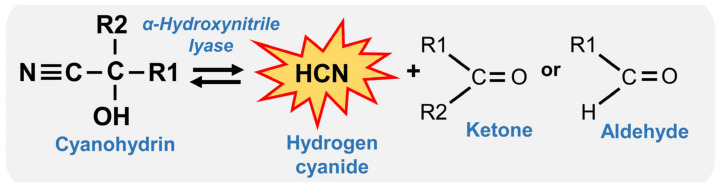
Central core of plant cyanogenesis. Interconversion of cyanohydrins (α-hydroxynitriles) into hydrogen cyanide (HCN) and ketone or aldheyde catalyzed by the α-hydroxynitrile lyase.

**Figure 2 plants-13-01239-f002:**
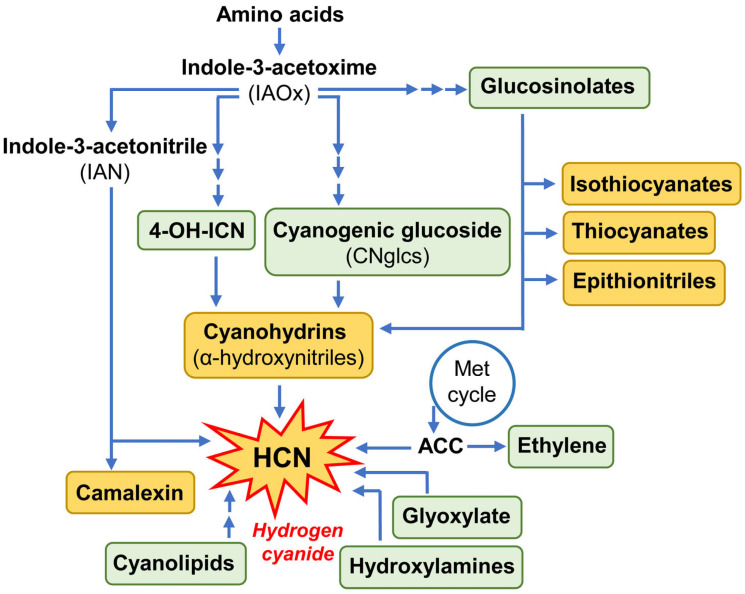
Scheme of the metabolic crosstalk of plant pathways involved in HCN production. Green boxes show plant compounds associated with HCN synthesis. Orange boxes indicate metabolites with defense properties against herbivores.

**Figure 3 plants-13-01239-f003:**
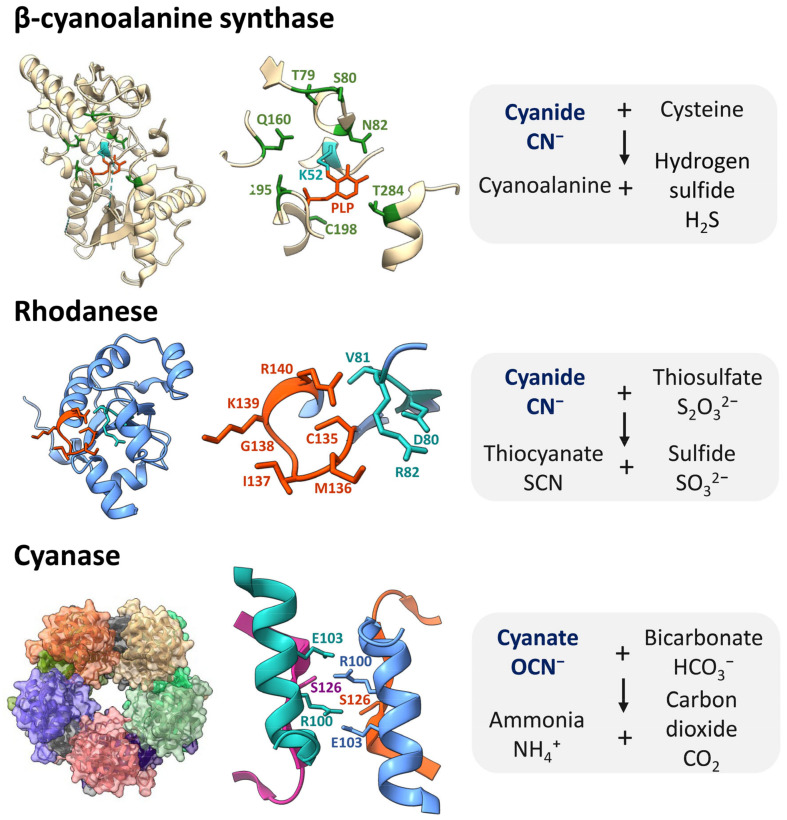
Enzymes for cyanide detoxification in arthropods. Tridimensional structures of β-cyanoalanine synthase (PDB code: 6XO2) and cyanase from *T. urticae* (PDB code: 5UK3), and rhodanase from *P. rapae* (AlphaFold code: AF-A0A345BJE4-F1). Ribbon representations and detailed views of the reactive sites are shown. For the β-cyanoalanine synthase, amino acids forming hydrogen bonds and covalent interaction with the pyridoxal phosphate (PLP) cofactor are coloured green and blue, respectively. The rhodanese conserved helix containing the reactive Cys and the stabilizing DVR conserved motif are coloured orange and blue, respectively. For the cyanase, the decameric structure and the dimeric reactive site formed by residues from four different monomers are monomer-specific coloured. The reactions catalysed by these enzymes are displayed.

**Figure 4 plants-13-01239-f004:**
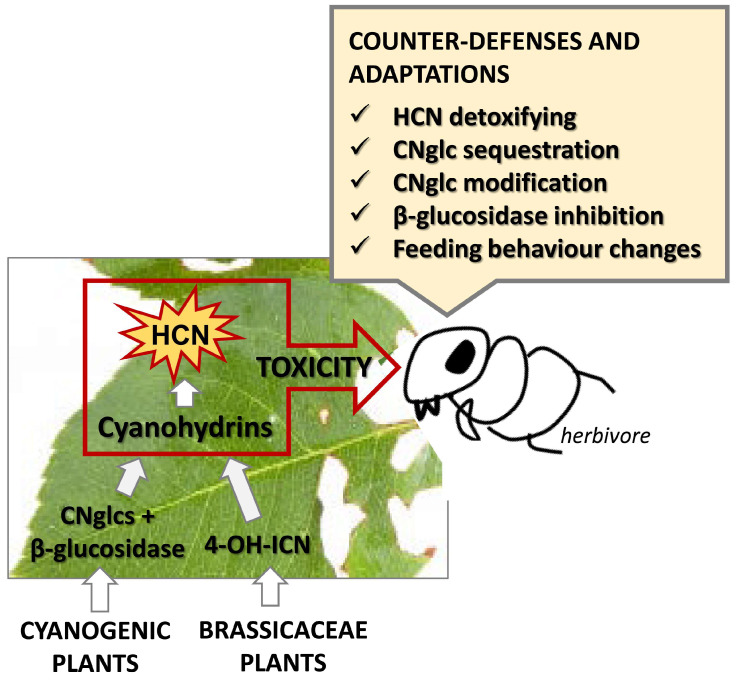
Schematic diagram representing a summary of the events associated with cyanide defences in the plant-herbivore context.

## Data Availability

Not applicable.

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
