# Peer review of "Plant Cyanogenic-Derived Metabolites and Herbivore Counter-Defences"

_plants, 2024, doi:10.3390/plants13091239_

Round 1
Reviewer 1 Report
Comments and Suggestions for Authors
Summary Comment:
This review provides a comprehensive overview of the complex interactions between plants and herbivores mediated by cyanogenic compounds. The authors thoroughly describe the biosynthesis, regulation and defensive functions of cyanogenic glycosides and other cyanide-releasing metabolites in plants. The review is well-structured with informative figures. With some additional ecological and evolutionary framing, and discussion of future research directions, this will be a valuable and timely synthesis.
Suggestions:
1. More background on the phylogenetic and ecological distribution of cyanogenic defenses in plants would be useful. Which taxa and growth forms most commonly employ them? Are there associations with particular habitats or herbivore pressures?
2. The evolutionary forces shaping cyanogenesis and herbivore counter-adaptations could be discussed further. What are the selection pressures and trade-offs for plants and herbivores? Is there evidence of coevolution, adaptation, and counter-adaptation? Bringing in concepts like the "mustard oil bomb" and "co-evolutionary arms race" could add interesting perspectives.
3. Quantitative estimates of the effectiveness of different herbivore counter-defense strategies would be informative. Which behavioral, physiological and biochemical adaptations contribute most to herbivore resistance in different taxa and feeding guilds? Are particular combinations of strategies synergistic?
4. The costs and benefits of cyanogenic defenses for plants could be considered more. How do resource allocation, autotoxicity and other factors constrain cyanogenesis?
5. The discussion of β-cyanoalanine synthase and rhodanese detoxification enzymes in herbivores could note their occurrence in other animal taxa. Homologs involved in cyanide metabolism have been characterized in some insects, nematodes and other groups, suggesting deep evolutionary roots.
6. Future directions for research on the mechanisms, ecology and evolution of cyanogenic plant-herbivore interactions could be elaborated. Key unresolved questions include: How is cyanogenesis regulated in response to herbivory and other stresses?
Author Response
Dear reviewer,
Thank you very much for your revision. We appreciate very much your efforts and indications to improve the manuscript quality. Our responses are indicated below your comments.
This review provides a comprehensive overview of the complex interactions between plants and herbivores mediated by cyanogenic compounds. The authors thoroughly describe the biosynthesis, regulation and defensive functions of cyanogenic glycosides and other cyanide-releasing metabolites in plants. The review is well-structured with informative figures. With some additional ecological and evolutionary framing, and discussion of future research directions, this will be a valuable and timely synthesis.
Thank you very much for your corrections and comments. Your suggestions are great and have pushed us to clarify some points and enhance the manuscript content.
Suggestions:
1.More background on the phylogenetic and ecological distribution of cyanogenic defenses in plants would be useful. Which taxa and growth forms most commonly employ them? Are there associations with particular habitats or herbivore pressures?
Despite evolutionary relevance, scarce attempts have been made to understand why some plant species are cyanogenic and what are the particular pressures associated with the acquisition of this feature. We have included a paragraph in the 2.1 section regarding this issue.
2.The evolutionary forces shaping cyanogenesis and herbivore counter-adaptations could be discussed further. What are the selection pressures and trade-offs for plants and herbivores? Is there evidence of coevolution, adaptation, and counter-adaptation? Bringing in concepts like the "mustard oil bomb" and "co-evolutionary arms race" could add interesting perspectives.
Again, scarce information exists on the coevolution, adaptation, and counter-adaptation of plants and herbivores. We have included a paragraph in the 4.1 section regarding this issue.
3.Quantitative estimates of the effectiveness of different herbivore counter-defense strategies would be informative. Which behavioral, physiological and biochemical adaptations contribute most to herbivore resistance in different taxa and feeding guilds? Are particular combinations of strategies synergistic?
Although this is a relevant point, no attempts have been made to determine the best herbivore counter-defense strategy. It depends so much on the specific plant-herbivore interaction and probably on the feeding guild. Further studies are needed to enable a solid discussion on the specificities and synergies in particular plant-herbivore interactions.
4.The costs and benefits of cyanogenic defenses for plants could be considered more. How do resource allocation, autotoxicity and other factors constrain cyanogenesis?
We have added a paragraph in the 2.1. section regarding this issue.
5.The discussion of β-cyanoalanine synthase and rhodanese detoxification enzymes in herbivores could note their occurrence in other animal taxa. Homologs involved in cyanide metabolism have been characterized in some insects, nematodes and other groups, suggesting deep evolutionary roots.
We have added two sentences in the 4.2.1 and 4.2.2 sections discussing the evolutionary origin of these enzymes in herbivores.
6.Future directions for research on the mechanisms, ecology and evolution of cyanogenic plant-herbivore interactions could be elaborated. Key unresolved questions include: How is cyanogenesis regulated in response to herbivory and other stresses.
Regarding this point, we consider that some of the suggestions indicated by the reviewer are already mentioned in the conclusion section. However, to complete this issue, some sentences have been added at the end of this section.

Reviewer 2 Report
Comments and Suggestions for Authors
This is an interesting review paper summarizing the state of knowledge about cyanogenic precursors, cyanide, cyanogenesis and its role in herbivore defense. The authors used important literature, including the most recent ones from recent years. I recommend the paper for publication in its current form after correcting a few inaccuracies and typos:
Keywords: please re-write keyworda taking care not to repeat words and phrases contained in the title of the paper
Line 90: Sorgum, Prunus - should be italicized
Line 110: Arabidopsis - should be italicized
Line 173: as above
Line 438, References, no 13 – year of publication should be 2010
Author Response
Dear reviewer,
Thank you very much for your revision. We appreciate very much your efforts and indications to improve the manuscript quality. Our responses are indicated below your comments.
Interesting review paper summarizing the state of knowledge about cyanogenic precursors, cyanide, cyanogenesis and its role in herbivore defense. The authors used important literature, including the most recent ones from recent years. I recommend the paper for publication in its current form after correcting a few inaccuracies and typos:
Keywords: please re-write keyworda taking care not to repeat words and phrases contained in the title of the paper
Line 90: Sorgum, Prunus - should be italicized
Line 110: Arabidopsis - should be italicized
Line 173: as above
Line 438, References, no 13 – year of publication should be 2010
We appreciate very much reviewer´s comments. Keywords have been modified and corrections have been made according to your indications.

Reviewer 3 Report
Comments and Suggestions for Authors
The submitted manuscript “Plant Cyanogenic-Derived Metabolites and Herbivore Counter-Defences” is a short review of the current knowledge on cyanogenesis in plants and mechanisms in herbivores to cope with these toxins. The authors collected the available literature on the subject and arranged the material into several paragraphs. The structure of the review is, in my opinion, correct and covers main issues of the subject. Overall, the review is valuable and provides sufficient amount of references for the reader to find more details not included in the review which must be synthetic as a rule. Nevertheless, the Authors did not avoid a number of inconsistencies, which I described below.
The text includes a lot of anthropomorphisms and teleological implications. I suggest the rephrasing of the statements, like those: ‘plants sense’…’plants react’… e.t.c. (lines 39, 43), ‘Brassicaceae has developed…’ (150), ‘they improved their capability to defend…’ (185), ‘This fact has pushed arthropods to design strategies along the evolution’ (240). You described the evolutionary ‘arms race’, but it should have been depicted using the evolutionary nomenclature.
‘The two-step mechanism has been developed to avoid autotoxicity’ (line 56) – again, this statement implies teleology, the purposefulness of evolution.
47: ‘Against pests, pathogens, and even animals’ – what do you mean by ‘pests’? ‘Pest’ is a very broad idea - lots of pests are animals, insects are animals and pathogens are pests; I suggest to rephrase the statement.
90, 110, 135, 173, 181, 191, 319, 350: please use italics for Latin generic names
148: choose singular or plural in this heading, e.g., Cyanohydrins: A convergent molecule….
168: should be: Brassicaceae
191: should be: They have mainly been studied
203: should be: all these metabolites having toxic
207-217: this passage should be rewritten and developed with more details on arthropods, especially insects, providing a more detailed knowledge on the mechanism of HCN action; the toxicity of HCN involving hemoglobin relates only to animals using hemoglobin for respiration – like in humans; in insects, oxygen is transferred directly to the tissues using the tracheal system.
243: should be: phloem sap
271: should be: their digestive tract
276: should be: of this lepidopteran
Figure 3: Is this the original figure or reproduced from a previously published material? Please, cite the source, if the images (or some of them) come from other sources.
Comments on the Quality of English Language
Please, check the English, especially the word order in sentences.
Author Response
Dear reviewer,
Thank you very much for your revision. We appreciate very much your efforts and indications to improve the manuscript quality. Our responses are indicated below your comments.
The submitted manuscript “Plant Cyanogenic-Derived Metabolites and Herbivore Counter-Defences” is a short review of the current knowledge on cyanogenesis in plants and mechanisms in herbivores to cope with these toxins. The authors collected the available literature on the subject and arranged the material into several paragraphs. The structure of the review is, in my opinion, correct and covers main issues of the subject. Overall, the review is valuable and provides sufficient amount of references for the reader to find more details not included in the review which must be synthetic as a rule. Nevertheless, the Authors did not avoid a number of inconsistencies, which I described below.
Thank you very much for your corrections. Your indications help to improve the review content.
The text includes a lot of anthropomorphisms and teleological implications. I suggest the rephrasing of the statements, like those: ‘plants sense’…’plants react’… e.t.c. (lines 39, 43), ‘Brassicaceae has developed…’ (150), ‘they improved their capability to defend…’ (185), ‘This fact has pushed arthropods to design strategies along the evolution’ (240). You described the evolutionary ‘arms race’, but it should have been depicted using the evolutionary nomenclature.
‘The two-step mechanism has been developed to avoid autotoxicity’ (line 56) – again, this statement implies teleology, the purposefulness of evolution.
The reviewer is right and probably along the text there are some anthropomorphisms or teleology in terms of the purpose, but we consider that the manuscript content (which is a review and not a research article) is not ambiguous and is quite correct. However, to avoid imprecise phrasing, we have modified some of the sentences mentioned by the reviewer.
-47: ‘Against pests, pathogens, and even animals’ – what do you mean by ‘pests’? ‘Pest’ is a very broad idea - lots of pests are animals, insects are animals and pathogens are pests; I suggest to rephrase the statement.
Following the reviewer´s comments, the terms ‘pests and animals’ have been replaced by the term herbivores, mammals and birds.
-90, 110, 135, 173, 181, 191, 319, 350: please use italics for Latin generic names.
-148: choose singular or plural in this heading, e.g., Cyanohydrins: A convergent molecule….
-168: should be: Brassicaceae
-191: should be: They have mainly been studied.
-203: should be: all these metabolites having toxic.
All errors and mistakes have been corrected according to reviewer indications.
-207-217: this passage should be rewritten and developed with more details on arthropods, especially insects, providing a more detailed knowledge on the mechanism of HCN action; the toxicity of HCN involving hemoglobin relates only to animals using hemoglobin for respiration – like in humans; in insects, oxygen is transferred directly to the tissues using the tracheal system.
We appreciate very much this correction. The paragraph has been rewritten according to the reviewer´s indications.
-243: should be: phloem sap.
-271: should be: their digestive tract.
-276: should be: of this lepidopteran
All corrections have been done.
-Figure 3: Is this the original figure or reproduced from a previously published material? Please, cite the source, if the images (or some of them) come from other sources.
Figure 3 is original. All parts of this figure have been specifically designed for this publication. It has been made by the authors and has not been previously published.
-Comments on the Quality of English Language
Please, check the English, especially the word order in sentences.
The English has been corrected.
Reviewer 4 Report
Comments and Suggestions for Authors
This is a reasonably complete review of cyanogensis in plants and a somewhat less complete review of mechanisms for coping with cyanogenesis in herbivorous insects. The coverage is representative, fair, and balanced. It would have been useful to cite other reviews and mention what additional information and perspectives are presented by the current review. I have flagged some minor errors, and made suggestions for two additional citations.
Figure 1. Aldehyde is spelled incorrectly.
Genus names such as Arabidopsis and Prunus should be italicised
line 220. Cyanide interferes with
line 239-243. The feeding mode of spider mites and aphids indeed enables them to avoid activating the cyanogenesis defense, but it probably evolved before cyanogenesis was evolved by plants. The authors' inference that the piercing/sucking feeding mode evolved from some other feeding mode, as an evolutionary response to cyanogenesis in plants, is not correct.
On the other hand, it is probable that leaf-snipping behavior by lepidoptera did evolve in response to cyanogenesis, or to other chemical defense mechanisms of plants.
line 274. surprising
line 278. Please cite the following experimental study of the insensitivity of S. eridania to cyanide. This is the most recent one I could find, but at least it is an experimental study that backs up the hypothesis stated (without citation) by the authors.
Heisler, Charles R., William F. Hodnick, and Sami Ahmad. 1988. Evidence for target site insensitivity to cyanide in Spodoptera eridania larvae. Comparative Biochemistry and Physiology Part C: Comparative Pharmacology 91(2): 469-472.
line 229. The section on herbivore counter-defences is incomplete, even though it reviews some very interesting recent studies in mites. Please cite the following review, which although out of date, contains more details on herbivore counter-defense strategies. Either cite this review, or the authors should update Section 3.3 of that review by citing relevant articles discussed as well as more recent ones.
Heckel, David G. 2014. Insect detoxification and sequestration strategies. Annual Plant Reviews 47: 77-114
Author Response
Dear reviewer,
Thank you very much for your revision. We appreciate very much your efforts and indications to improve the manuscript quality. Our responses are indicated below your comments.
This is a reasonably complete review of cyanogensis in plants and a somewhat less complete review of mechanisms for coping with cyanogenesis in herbivorous insects. The coverage is representative, fair, and balanced. It would have been useful to cite other reviews and mention what additional information and perspectives are presented by the current review. I have flagged some minor errors, and made suggestions for two additional citations.
Thank you very much for your corrections, comments and indications.
-Figure 1. Aldehyde is spelled incorrectly.
The reviewer is right. Thank you. The word has been now correctly spelled.
-Genus names such as Arabidopsis and Prunus should be italicised.
-line 220. Cyanide interferes with
Corrections have been done
-line 239-243. The feeding mode of spider mites and aphids indeed enables them to avoid activating the cyanogenesis defense, but it probably evolved before cyanogenesis was evolved by plants. The authors' inference that the piercing/sucking feeding mode evolved from some other feeding mode, as an evolutionary response to cyanogenesis in plants, is not correct. On the other hand, it is probable that leaf-snipping behavior by lepidoptera did evolve in response to cyanogenesis, or to other chemical defense mechanisms of plants.
The reviewer´s comment is right. Our purpose was not to indicate that piercing/sucking feeding mode evolved from some other feeding modes as an evolutionary response to cyanogenesis in plants. Our aim was to mention that phytophagous arthropods try to avoid plant defensive compounds, including those derived from cyanogenesis. In this context, limiting tissue disruption seems to be an appropriate alternative to prevent cyanide production. Thus, alterations in their feeding behaviour can help for this purpose. The text has been modified to avoid this misunderstanding.
-line 274. Surprising
The word surprising has been deleted.
-line 278. Please cite the following experimental study of the insensitivity of S. eridania to cyanide. This is the most recent one I could find, but at least it is an experimental study that backs up the hypothesis stated (without citation) by the authors.
Heisler, Charles R., William F. Hodnick, and Sami Ahmad. 1988. Evidence for target site insensitivity to cyanide in Spodoptera eridania larvae. Comparative Biochemistry and Physiology -Part C: Comparative Pharmacology 91(2): 469-472.
We appreciate very much the reviewer´s indications and the recommended reference. The manuscript published by Heisler et al. (1988) has been added as well as some results included in.
-line 229. The section on herbivore counter-defences is incomplete, even though it reviews some very interesting recent studies in mites. Please cite the following review, which although out of date, contains more details on herbivore counter-defense strategies. Either cite this review, or the authors should update Section 3.3 of that review by citing relevant articles discussed as well as more recent ones.
Heckel, David G. 2014. Insect detoxification and sequestration strategies. Annual Plant Reviews 47: 77-114
According to reviewer´s indication, the mentioned reference has been added. This excellent publication compiles different insect detoxification and sequestration strategies to fight against the plethora of plant defensive compounds. The section 3.3 is particularly focused on strategies to control cyanogenic glucoside derivatives. Since our review is dedicated to insect counter-defences to overcome cyanide and cyanohydrin production, the reference has been included and additional comments have been incorporated.
Round 2
Reviewer 1 Report
Comments and Suggestions for Authors
The authors have added some discussion on the evolutionary aspects of cyanogenesis, mentioning selection pressures like soil nitrogen levels that may drive the gain/loss of cyanogenic capabilities. This helps address the previous weakness of limited evolutionary context. But the authors could do some improvements.
1. While the authors discuss the toxicity and modes of action in more detail, specific concentrations required for deterrence/toxicity are still lacking where such data exists.
2. The scope remains limited to vascular plants, with no added discussion of cyanogenesis in algae, bryophytes or other early-diverging lineages to provide evolutionary context. This weakness persists.
3. Some additional molecular details have been added, such as the CYP79 and CYP71 enzymes involved in cyanogenic glycoside biosynthesis. However, discussion of signaling and regulatory pathways controlling cyanogenesis is still rather limited.
4. The authors have not added any overarching summary diagram or conceptual framework to help synthesize the major concepts and findings. This could still improve the accessibility of the key insights.
In summary, several key shortcomings remain, including the lack of quantitative defensive data, limited molecular regulatory details, missing ecological and agricultural perspectives, and a unifying visual model or future directions section.
Author Response
The authors have added some discussion on the evolutionary aspects of cyanogenesis, mentioning selection pressures like soil nitrogen levels that may drive the gain/loss of cyanogenic capabilities. This helps address the previous weakness of limited evolutionary context. But the authors could do some improvements.
Thank you. We appreciate your comments.
- While the authors discuss the toxicity and modes of action in more detail, specific concentrations required for deterrence/toxicity are still lacking where such data exists.
The required information has been added in section 3.
- The scope remains limited to vascular plants, with no added discussion of cyanogenesis in algae, bryophytes or other early-diverging lineages to provide evolutionary context. This weakness persists.
We have broadened the evolutionary scope. Some sentences have been added in the first paragraph of section 2.1.
- Some additional molecular details have been added, such as the CYP79 and CYP71 enzymes involved in cyanogenic glycoside biosynthesis. However, discussion of signaling and regulatory pathways controlling cyanogenesis is still rather limited.
Following reviewer´s comments, an additional paragraph has been added in section 2.1.
- The authors have not added any overarching summary diagram or conceptual framework to help synthesize the major concepts and findings. This could still improve the accessibility of the key insights.
A summary diagram has been added in the conclusion section according to the reviewer´s indications.
In summary, several key shortcomings remain, including the lack of quantitative defensive data, limited molecular regulatory details, missing ecological and agricultural perspectives, and a unifying visual model or future directions section.
